# ADJUSTABLE REAL-TIME STYLE TRANSFER

## ABSTRACT

Artistic style transfer is the problem of synthesizing an image with content similar to a given image and style similar to another. Although recent feed-forward neural networks can generate stylized images in real-time, these models produce a single stylization given a pair of style/content images, and the user doesn't have control over the synthesized output. Moreover, the style transfer depends on the hyper-parameters of the model with varying "optimum" for different input images. Therefore, if the stylized output is not appealing to the user, she/he has to try multiple models or retrain one with different hyper-parameters to get a favorite stylization. In this paper, we address these issues by proposing a novel method which allows adjustment of crucial hyper-parameters, after the training and in real-time, through a set of manually adjustable parameters. These parameters enable the user to modify the synthesized outputs from the same pair of style/content images, in search of a favorite stylized image. Our quantitative and qualitative experiments indicate how adjusting these parameters is comparable to retraining the model with different hyper-parameters. We also demonstrate how these parameters can be randomized to generate results which are diverse but still very similar in style and content.

## 1 INTRODUCTION

Style transfer is a long-standing problem in computer vision with the goal of synthesizing new images by combining the *content* of one image with the *style* of another (Efros & Freeman, 2001; Hertzmann, 1998; Ashikhmin, 2001). Recently, neural style transfer techniques (Gatys et al., 2015; 2016; Johnson et al., 2016; Ghiasi et al., 2017; Li et al., 2018; 2017b) showed that the correlation between the features extracted from the trained deep neural networks is quite effective on capturing the visual styles and content that can be used for generating images *similar* in style and content. However, since the definition of similarity is inherently vague, the objective of style transfer is not well defined (Dumoulin et al., 2017) and one can imagine multiple stylized images from the same pair of content/style images.

Existing real-time style transfer methods generate only one stylization for a given content/style pair and while the stylizations of different methods usually look distinct (Sanakoyeu et al., 2018; Huang & Belongie, 2017), it is not possible to say that one stylization is better in all contexts since people react differently to images based on their background and situation. Hence, to get favored stylizations users must try different methods that is not satisfactory. It is more desirable to have a single model which can generate *diverse* results, but still *similar* in style and content, in real-time, by adjusting some input parameters.

One other issue with the current methods is their high sensitivity to the hyper-parameters. More specifically, current real-time style transfer methods minimize a weighted sum of losses from different layers of a pre-trained image classification model (Johnson et al., 2016; Huang & Belongie, 2017) (check Sec 3 for details) and different weight sets can result into very different styles (Figure 6). However, one can only observe the effect of these weights in the final stylization by fully retraining the model with the new set of weights. Considering the fact that the "optimal" set of weights can be different for any pair of style/content (Figure 3) and also the fact that this "optimal" truly doesn't exist (since the goodness of the output is a personal choice) retraining the models over and over until the desired result is generated is not practical.

The primary goal of this paper is to address these issues by providing a novel mechanism which allows for adjustment of the stylized image, in ***real-time*** and ***after*** training. To achieve this, we use an auxiliary network which accepts additional parameters as inputs and changes the style transfer process by adjusting the weights between multiple losses. We show that changing these parameters at inference time results to stylizations similar to the ones achievable by retraining the model with different hyperparameters. We also show that a random selection of these parameters at run-time can generate a random stylization. These solutions, enable the end user to be in full control of how the stylized image is being formed as well as having the capability of generating multiple stochastic stylized images from a fixed pair of style/content. The stochastic nature of our proposed method is most apparent when viewing the transition between random generations. Therefore, we highly encourage the reader to check the project website https://goo.gl/PVWQ9K to view the generated stylizations.

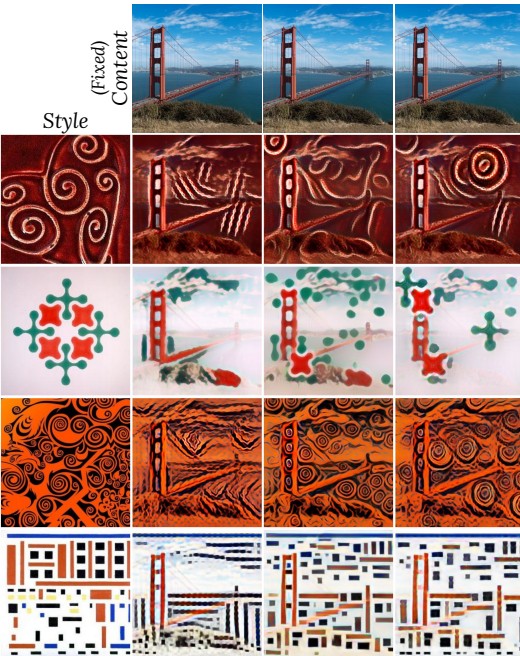

Figure 1: Adjusting the output of the synthesized stylized images in real-time. Each column shows a different stylized image for the same content and style image. Note how each row still resembles the same content and style while being widely different in details.

## 2 RELATED WORK

The strength of deep networks in style transfer was first demonstrated by Gatys et al. (Gatys et al., 2016). While this method generates impressive results, it is too slow for real-time applications due to its optimization loop. Follow up works speed up this process by training feed-forward networks that can transfer style of a single style image (Johnson et al., 2016; Ulyanov et al., 2016) or multiple styles (Dumoulin et al., 2017). Other works introduced real-time methods to transfer style of arbitrary style image to an arbitrary content image (Ghiasi et al., 2017; Huang & Belongie, 2017). These methods can generate different stylizations from different style images; however, they only produce one stylization for a single pair of content/style image which is different from our proposed method.

Generating diverse results have been studied in multiple domains such as colorizations (Deshpande et al., 2017; Cao et al., 2017), image synthesis (Chen & Koltun, 2017), video prediction (Babaeizadeh et al., 2017; Lee et al., 2018), and domain transfer (Huang et al., 2018; Zhang, 2018). Domain transfer is the most similar problem to the style transfer. Although we can generate multiple outputs from a given input image (Huang et al., 2018), we need a collection of target or style images for training. Therefore we can not use it when we do not have a collection of similar styles.

Style loss function is a crucial part of style transfer which affects the output stylization significantly. The most common style loss is Gram matrix which computes the second-order statistics of the feature activations (Gatys et al., 2016), however many alternative losses have been introduced to measure distances between feature statistics of the style and stylized images such as correlation alignment loss (Peng & Saenko, 2018), histogram loss (Risser et al., 2017), and MMD loss (Li et al., 2017a). More recent work (Liu et al., 2017) has used depth similarity of style and stylized images as a part of the loss. We demonstrate the success of our method using only Gram matrix; however, our approach can be expanded to utilize other losses as well.

To the best of our knowledge, the closest work to this paper is (Ulyanov et al., 2017) in which the authors utilized Julesz ensemble to encourage diversity in stylizations explicitly. Although this

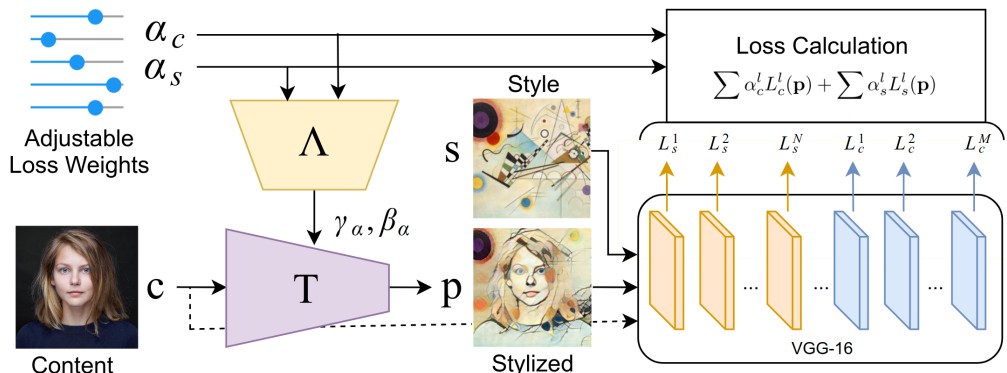

Figure 2: Architecture of the proposed model. The loss adjustment parameters $\boldsymbol{\alpha}_c$ and $\boldsymbol{\alpha}_s$ is passed to the network $\Lambda$ which will predict activation normalizers $\gamma_{\boldsymbol{\alpha}}$ and $\beta_{\boldsymbol{\alpha}}$ that normalize activation of main stylizing network $T$. The stylized image is passed to a trained image classifier where its intermediate representation is used to calculate the style loss $\mathcal{L}_s$ and content loss $\mathcal{L}_c$. Then the loss from each layer is multiplied by the corresponding input adjustment parameter. Models $\Lambda$ and $T$ are trained jointly by minimizing this weighted sum. At generation time, values for $\boldsymbol{\alpha}_c$ and $\boldsymbol{\alpha}_s$ can be adjusted manually or randomly sampled to generate varied stylizations.

method generates different stylizations, they are very similar in style, and they only differ in minor details. A qualitative comparison in Figure 8 shows that our proposed method is more effective in diverse stylization.

## 3 BACKGROUND

### 3.1 STYLE TRANSFER USING DEEP NETWORKS

Style transfer can be formulated as generating a stylized image $\mathbf{p}$ which its content is similar to a given content image $\mathbf{c}$ and its style is close to another given style image $\mathbf{s}$.

$$\mathbf{p} = \Psi(\mathbf{c}, \mathbf{s})$$

The similarity in style can be vaguely defined as sharing the same spatial statistics in low-level features, while similarity in content is roughly having a close Euclidean distance in high-level features (Ghiasi et al., 2017). These features are typically extracted from a pre-trained image classification network, commonly VGG-19 (Simonyan & Zisserman, 2014). The main idea here is that the features obtained by the image classifier contain information about the content of the input image while the correlation between these features represents its style.

In order to increase the similarity between two images, Gatys et al. (Gatys et al., 2016) minimize the following distances between their extracted features:

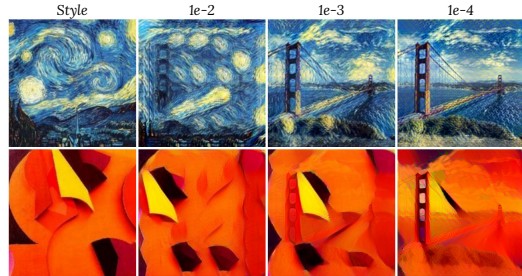

Figure 3: Effect of adjusting the style weight in style transfer network from (Johnson et al., 2016). Each column demonstrates the result of a separate training with all $w_s^l$ set to the printed value. As can be seen, the "optimal" weight is different from one style image to another and there can be multiple "good" stylizations depending on ones' personal choice. Check supplementary materials for more examples.

$$\mathcal{L}_c^l(\mathbf{p}) = \left|\left|\phi^l(\mathbf{p}) - \phi^l(\mathbf{s})\right|\right|_2^2, \qquad \mathcal{L}_s^l(\mathbf{p}) = \left|\left|G(\phi^l(\mathbf{p})) - G(\phi^l(\mathbf{s}))\right|\right|_F^2 \qquad (1)$$

where $\phi^l(\mathbf{x})$ is activation of a pre-trained classification network at layer $l$ given the input image $\mathbf{x}$, while $\mathcal{L}_c^l(\mathbf{p})$ and $\mathcal{L}_s^l(\mathbf{p})$ are content and style loss at layer $l$ respectively. $G(\phi^l(\mathbf{p}))$ denotes the Gram matrix associated with $\phi^l(\mathbf{p})$.

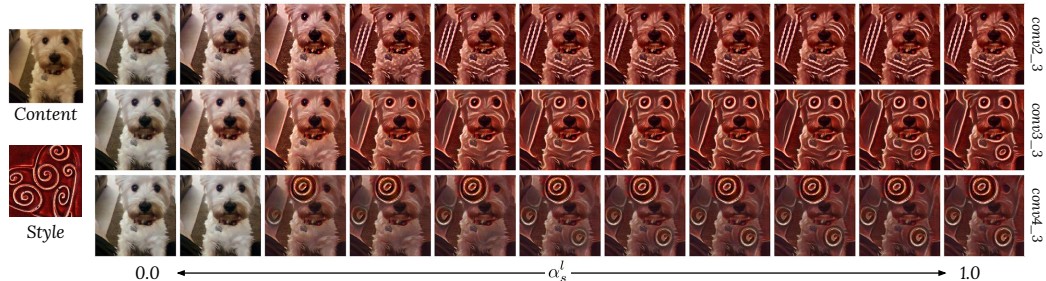

Figure 4: Effect of adjusting the input parameters $\boldsymbol{\alpha}_s$ on stylization. Each row shows the stylized output when a single $\alpha_s^l$ increased gradually from zero to one while other $\boldsymbol{\alpha}_s$ are fixed to zero. Notice how the details of each stylization is different specially at the last column where the value is maximum. Also note how deeper layers use *bigger* features of style image to stylize the content.

The total loss is calculated as a weighted sum of losses across a set of *content layers* $C$ and *style layers* $S$:

$$\mathcal{L}_c(\mathbf{p}) = \sum_{l \in C} w_c^l \mathcal{L}_c^l(\mathbf{p}), \qquad \mathcal{L}_s(\mathbf{p}) = \sum_{l \in S} w_s^l \mathcal{L}_s^l(\mathbf{p}) \tag{2}$$

where $w_c^l$, $w_s^l$ are hyper-parameters to adjust the contribution of each layer to the loss. Layers can be shared between $C$ and $S$. These hyper-parameters have to be manually fine tuned through try and error and usually vary for different style images (Figure 3). Finally, the objective of style transfer can be defined as:

$$\min_{\mathbf{p}} \left( \mathcal{L}_c(\mathbf{p}) + \mathcal{L}_s(\mathbf{p}) \right) \tag{3}$$

This objective can be minimized by iterative gradient-based optimization methods starting from an initial $\mathbf{p}$ which usually is random noise or the content image itself.

## 3.2 REAL-TIME FEED-FORWARD STYLE TRANSFER

Solving the objective in Equation 3 using an iterative method can be very slow and has to be repeated for any given pair of style/content image. A much faster method is to directly train a deep network $T$ which maps a given content image $\mathbf{c}$ to a stylized image $\mathbf{p}$ (Johnson et al., 2016). $T$ is usually a feed-forward convolutional network (parameterized by $\theta$) with residual connections between down-sampling and up-sampling layers (Ruder et al., 2018) and is trained on many content images using Equation 3 as the loss function:

$$\min_{\theta} \left( \mathcal{L}_c(T(\mathbf{c})) + \mathcal{L}_s(T(\mathbf{c})) \right) \tag{4}$$

The style image is assumed to be fixed and therefore a different network should be trained per style image. However, for a fixed style image, this method can generate stylized images in real-time (Johnson et al., 2016). Recent methods (Dumoulin et al., 2017; Ghiasi et al., 2017; Huang & Belongie, 2017) introduced real-time style transfer methods for multiple styles. But, these methods still generate only one stylization for a pair of style and content images.

## 4 PROPOSED METHOD

## 4.1 PROBLEM STATEMENT

In this paper we address the following issues in real-time feed-forward style transfer methods:
1. The output of these models is sensitive to the hyper-parameters $w_c^l$ and $w_s^l$ and different weights significantly affect the generated stylized image as demonstrated in Figure 6. Moreover, the "optimal" weights vary from one style image to another (Figure 3) and therefore finding a good set of weights should be repeated for each style image. Please note that for each set of $w_c^l$ and $w_s^l$ the model has to be fully retrained that limits the practicality of style transfer models.

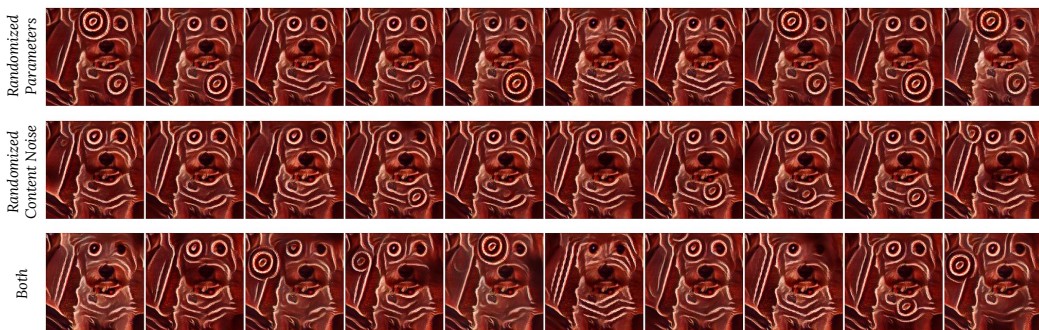

Figure 5: Effect of randomizing $\boldsymbol{\alpha}$ and additive Gaussian noise on stylization. Top row demonstrates that randomizing $\boldsymbol{\alpha}$ results to different stylizations however the style features appear in the same spatial position (e.g., look at the swirl effect on the left eye). Middle row visualizes the effect of adding random noise to the content image in moving these features with fixed $\boldsymbol{\alpha}$. Combination of these two randomization techniques can generate highly versatile outputs which can be seen in the bottom row. Notice how each image in this row differs in both style and the spatial position of style elements. Look at Figure 10 for more randomized results.

2. Current methods generate a *single* stylized image given a content/style pair. While the stylizations of different methods usually look very distinct (Sanakoyeu et al., 2018), it is not possible to say which stylization is better for every context since it is a matter of personal taste. To get a favored stylization, users may need to try different methods or train a network with different hyper-parameters which is not satisfactory and, ideally, the user should have the capability of getting different stylizations in real-time.

We address these issues by conditioning the generated stylized image on additional input parameters where each parameter controls the share of the loss from a corresponding layer. This solves the problem (1) since one can adjust the contribution of each layer to adjust the final stylized result after the training and in real-time. Secondly, we address the problem (2) by randomizing these parameters which result in different stylizations.

## 4.2 Style transfer with adjustable loss

We enable the users to adjust $w_c^l, w_s^l$ without retraining the model by replacing them with input parameters and conditioning the generated style images on these parameters:

$$\mathbf{p} = \Psi(\mathbf{c}, \mathbf{s}, \boldsymbol{\alpha}_c, \boldsymbol{\alpha}_s)$$

$\boldsymbol{\alpha}_c$ and $\boldsymbol{\alpha}_s$ are vectors of parameters where each element corresponds to a different layer in content layers $C$ and style layers $S$ respectively. $\alpha_c^l$ and $\alpha_s^l$ replace the hyper-parameters $w_c^l$ and $w_s^l$ in the objective Equation 2:

$$\mathcal{L}_c(\mathbf{p}) = \sum_{l \in C} \alpha_c^l \mathcal{L}_c^l(\mathbf{p}) \text{ and } \mathcal{L}_s(\mathbf{p}) \qquad = \sum_{l \in S} \alpha_s^l \mathcal{L}_s^l(\mathbf{p}) \qquad (5)$$

To learn the effect of $\boldsymbol{\alpha}_c$ and $\boldsymbol{\alpha}_s$ on the objective, we use a technique called *conditional instance normalization* (Ulyanov et al.). This method transforms the activations of a layer $x$ in the feed-forward network $T$ to a normalized activation $z$ which is conditioned on additional inputs $\boldsymbol{\alpha} = [\boldsymbol{\alpha}_c, \boldsymbol{\alpha}_s]$:

$$z = \gamma_{\boldsymbol{\alpha}}\left(\frac{x - \mu}{\sigma}\right) + \beta_{\boldsymbol{\alpha}} \qquad (6)$$

where $\mu$ and $\sigma$ are mean and standard deviation of activations at layer $x$ across spatial axes (Ghiasi et al., 2017) and $\gamma_{\boldsymbol{\alpha}}, \beta_{\boldsymbol{\alpha}}$ are the learned mean and standard deviation of this transformation. These parameters can be approximated using a second neural network which will be trained end-to-end with $T$:

$$\gamma_{\boldsymbol{\alpha}}, \beta_{\boldsymbol{\alpha}} = \Lambda(\boldsymbol{\alpha}_c, \boldsymbol{\alpha}_s) \qquad (7)$$

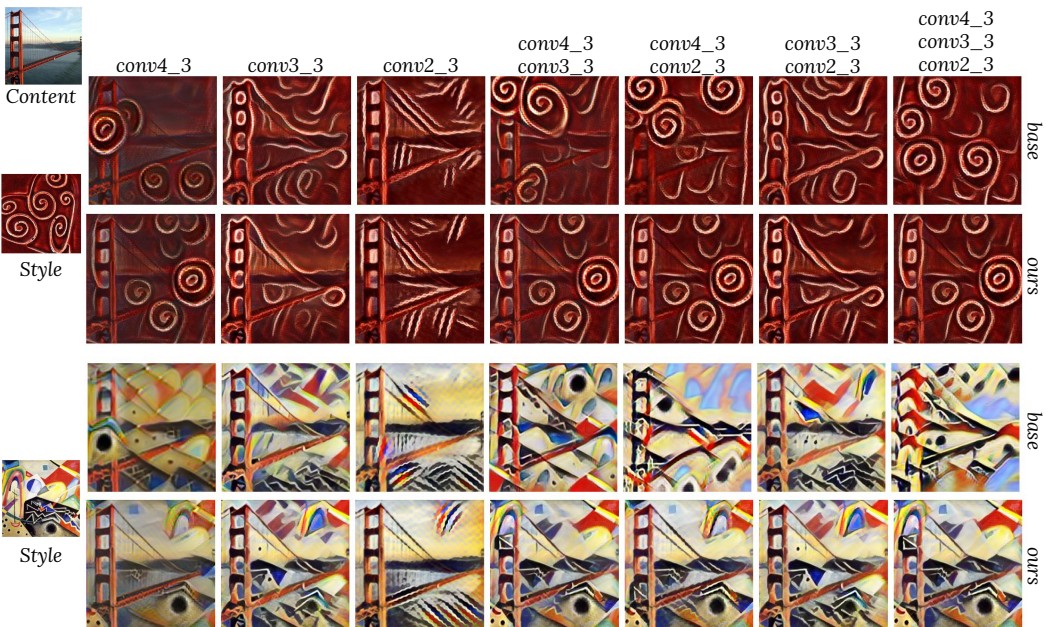

Figure 6: Qualitative comparison between the base model from (Johnson et al., 2016) with our proposed method. For the base model, each column has been retrained with all the weights set to zero except for the mentioned layers which has been set to $1\mathrm{e}-3$. For our model, the respective parameters $\alpha_s^l$ has been adjusted. Note how close the stylizations are and how the combination of layers stays the same in both models.

Since $\mathcal{L}^l$ can be very different in scale, one loss term may dominate the others which will fail the training. To balance the losses, we normalize them using their exponential moving average as a normalizing factor, i.e. each $\mathcal{L}^l$ will be normalized to:

$$\mathcal{L}^l(\mathbf{p}) = \frac{\sum_{i \in C \cup S} \overline{\mathcal{L}^i}(\mathbf{p})}{\overline{\mathcal{L}^l}(\mathbf{p})} * \mathcal{L}^l(\mathbf{p}) \tag{8}$$

where $\overline{\mathcal{L}^l}(\mathbf{p})$ is the exponential moving average of $\mathcal{L}^l(\mathbf{p})$.

## 5 EXPERIMENTS

In this section, first we study the effect of adjusting the input parameters in our method. Then we demonstrate that we can use our method to generate random stylizations and finally, we compare our method with a few baselines in terms of generating random stylizations.

### 5.1 IMPLEMENTATION DETAILS

We implemented $\Lambda$ as a multilayer fully connected neural network. We used the same architecture as (Johnson et al., 2016; Dumoulin et al., 2017; Ghiasi et al., 2017) for $T$ and only increased number of residual blocks by 3 (look at supplementary materials for details) which improved stylization results. We trained $T$ and $\Lambda$ jointly by sampling random values for $\boldsymbol{\alpha}$ from $U(0, 1)$. We trained our model on ImageNet (Deng et al., 2009) as content images while using paintings from Kaggle Painter by Numbers (Kaggle) and textures from Descibable Texture Dataset (Cimpoi et al., 2014) as style images. We selected random images form ImageNet test set, MS-COCO (Lin et al., 2014) and faces from CelebA dataset (Liu et al., 2018) as our content test images. Similar to (Ghiasi et al., 2017; Dumoulin et al., 2017), we used the last feature set of $conv3$ as content layer $C$. We used last feature set of $conv2$, $conv3$ and $conv4$ layers from VGG-19 network as style layers $S$. Since there is only one content layer, we fix $\boldsymbol{\alpha}_c = 1$. Our implementation can process $47.5$ fps on a NVIDIA GeForce 1080, compared to $52.0$ for the base model without $\Lambda$ sub-network.

## 5.2 Effect of adjusting the input parameters

The primary goal of introducing the adjustable parameters $\boldsymbol{\alpha}$ was to modify the loss of each separate layer manually. Qualitatively, this is demonstrable by increasing one of the input parameters from zero to one while fixing the rest of them to zero. Figure 4 shows one example of such transition. Each row in this figure is corresponding to a different style layer, and therefore the stylizations at each row would be different. Notice how deeper layers stylize the image with *bigger* stylization elements from the style image but all of them still apply the coloring. We also visualize the effect of increasing two of the input parameters at the same time in Figure 9. However, these transitions are best demonstrated interactively which is accessible at the project website https://goo.gl/PVWQ9K.

To quantitatively demonstrate the change in losses with adjustment of the input parameters, we rerun the same experiment of assigning a fixed value to all of the input parameters while gradually increasing one of them from zero to one, this time across 100 different content images. Then we calculate the median loss at each style loss layer $S$. As can be seen in Figure 7-(top), increasing $\alpha_s^l$ decreases the measured loss corresponding to that parameter. To show the generalization of our method across style images, we trained 25 models with different style images and then measured median of the loss at any of the $S$ layers for 100 different

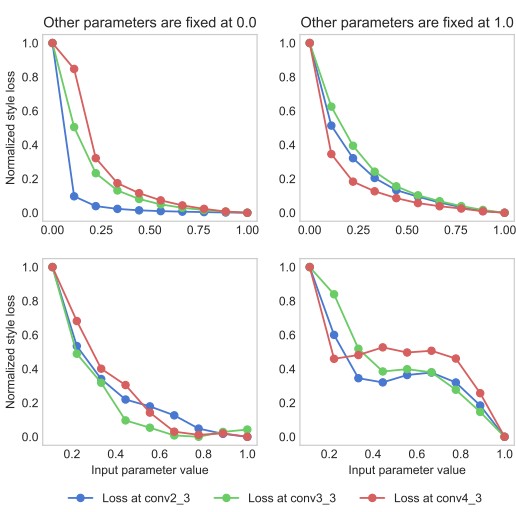

Figure 7: Effect of adjusting the input parameters $\boldsymbol{\alpha}_s$ on style loss from different layers across single style image of Figure 4 (top) or 25 different style images (bottom). In each curve, one of the input parameters $\alpha_s^l$ has been increased from zero to one while others are fixed at to zero (left) and to one (right). Then the style loss has been calculated across 100 different content images. As can be seen, increasing $\alpha_s^l$ decreases the loss of the corresponding layer. Note that the losses is normalized in each layer for better visualization.

content images (Figure 7)-(bottom). We exhibit the same drop trends as before which means the model can generate stylizations conditioned on the input parameters.

Finally, we verify that modifying the input parameters $\boldsymbol{\alpha}_s$ generates visually similar stylizations to the retrained base model with different loss weights $w_s^l$. To do so, we train the base model (Johnson et al., 2016) multiple times with different $w_s^l$ and then compare the generated results with the output of our model when $\forall l \in S, \alpha_s^l = w_s^l$. Figure 6 demonstrates this comparison. Note how the proposed stylizations in test time and without retraining match the output of the base model.

## 5.3 Generating randomized stylizations

One application of our proposed method is to generate multiple stylizations given a fixed pair of content/style image. To do so, we randomize $\boldsymbol{\alpha}$ to generate randomized stylization (top row of Figure 5). Changing values of $\boldsymbol{\alpha}$ usually do not randomize the position of the "elements" of the style. We can enforce this kind of randomness by adding some noise with the small magnitude to the content image. For this purpose, we multiply the content image with a mask which is computed by applying an inverse Gaussian filter on a white image with a handful ($< 10$) random zeros. This masking can shadow sensitive parts of the image which will change the spatial locations of the "elements" of style. Middle row in Figure 5 demonstrates the effect of this randomization. Finally, we combine these two randomizations to maximizes the diversity of the output which is shown in the bottom row of Figure 5. More randomized stylizations can be seen in Figure 10 and at https://goo.gl/PVWQ9K.

### 5.3.1 Comparison with other methods

To the best of our knowledge, generating diverse stylizations at real-time is only have been studied at (Ulyanov et al., 2017) before. In this section, we qualitatively compare our method with this baseline. Also, we compare our method with a simple baseline where we add noise to the style parameters.

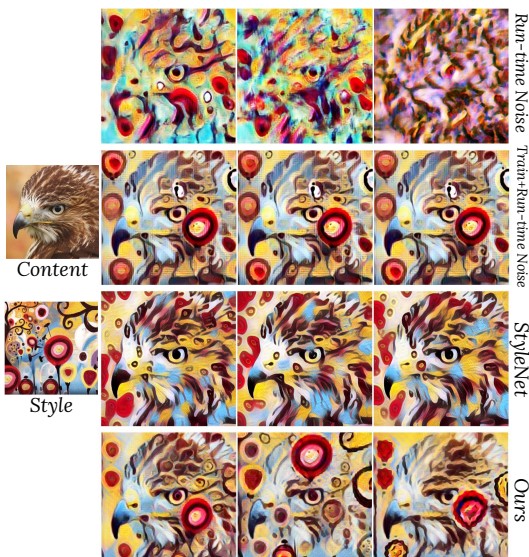

The simplest baseline for getting diverse stylizations is to add noises to some parameters or the inputs of the style-transfer network. In the last section, we demonstrate that we can move the locations of elements of style by adding noise to the content input image. To answer the question that if we can get different stylizations by adding noise to the style input of the network, we utilize the model of (Dumoulin et al., 2017) which uses conditional instance normalization for transferring style. We train this model with only one style image and to get different stylizations, we add random noise to the style parameters ($\gamma_{\boldsymbol{\alpha}}$ and $\beta_{\boldsymbol{\alpha}}$ parameters of equation 6) at run-time. The stylization results for this baseline are shown on the top row of Figure 8. While we get different stylizations by adding random noises, the stylizations are no longer similar to the input style image.

To enforce similar stylizations, we trained the same baseline while we add random noises at the training phase as well. The stylization results are shown in the second row of Figure 8. As it can be seen, adding noise at the training time makes the model robust to the noise and the stylization results are similar. This indicates that a loss term that encourages diversity is necessary.

Figure 8: Diversity comparison of our method and baselines. First row shows results for a baseline that adds random noises to the style parameters at run-time. While we get diverse stylizations, the results are not similar to the input style image. Second row contains results for a baseline that adds random noises to the style parameters at both training time and run-time. Model is robust to the noise and it does not generate diverse results. Third row shows stylization results of StyleNet (Ulyanov et al., 2017). Our method generates diverse stylizations while StyleNet results mostly differ in minor details.

We also compare the results of our model with StyleNet (Ulyanov et al., 2017). As visible in Figure 8, although StyleNet's stylizations are different, they vary in minor details and all carry the same level of stylization elements. In contrast, our model synthesizes stylized images with varying levels of stylization and more randomization.

## 6 CONCLUSION

Our main contribution in this paper is a novel method which allows adjustment of each loss layer's contribution in feed-forward style transfer networks, in real-time and after training. This capability allows the users to adjust the stylized output to find the favorite stylization by changing input parameters and without retraining the stylization model. We also show how randomizing these parameters plus some noise added to the content image can result in very different stylizations from the same pair of style/content image.

Our method can be expanded in numerous ways e.g. applying it to multi-style transfer methods such as (Dumoulin et al., 2017; Ghiasi et al., 2017), applying the same parametrization technique to randomize the correlation loss between *the features of each layer* and finally using different loss functions and pre-trained networks for computing the loss to randomize the outputs even further. One other interesting future direction is to apply the same "loss adjustment after training" technique for other classic computer vision and deep learning tasks. Style transfer is not the only task in which modifying the hyper-parameters can greatly affect the predicted results and it would be rather interesting to try this method for adjusting the hyper-parameters in similar problems.

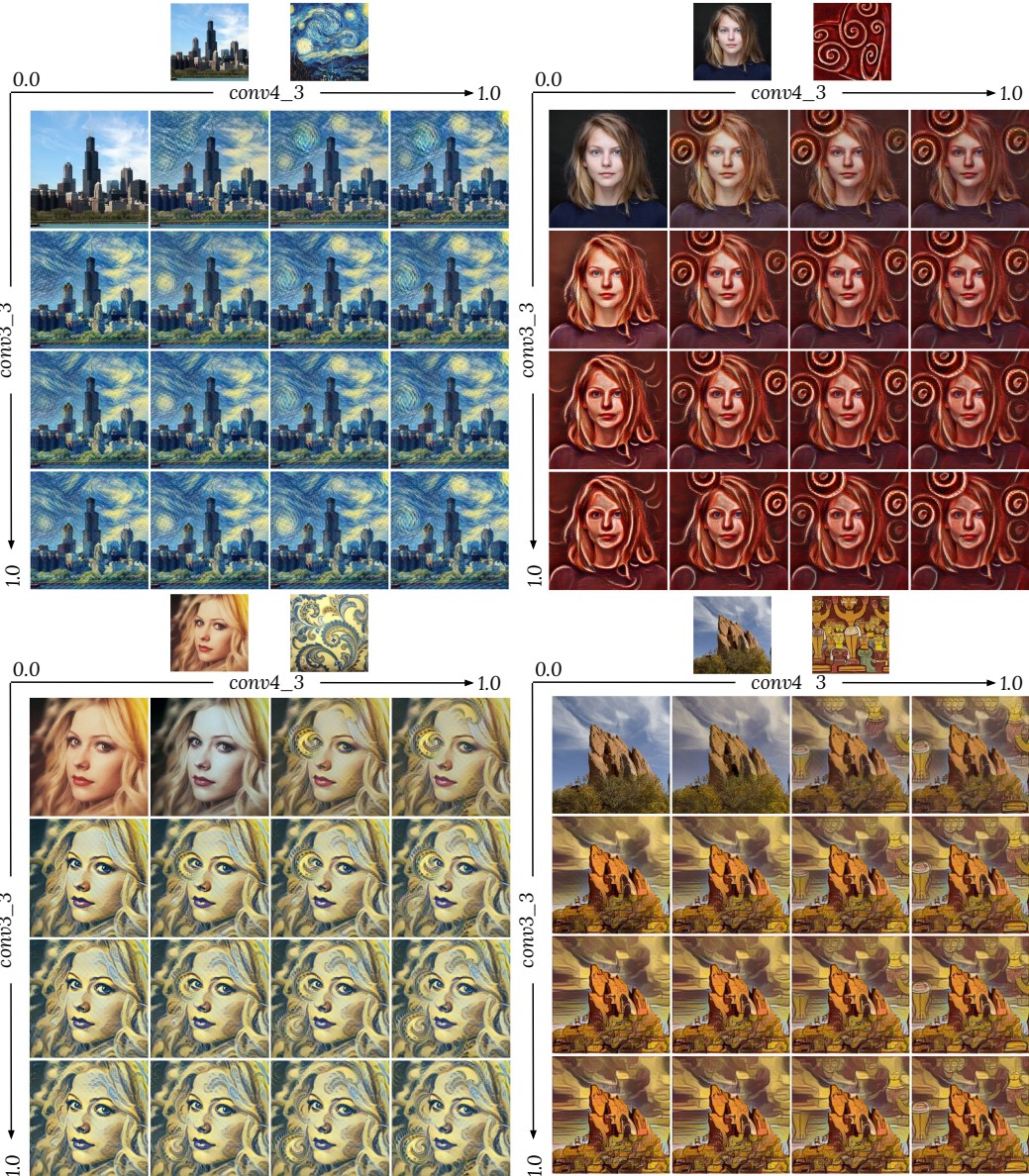

Figure 9: More results for adjusting the input parameters in real-time and after training. In each block the style/content pair is fixed while the parameters corresponding to $conv3$ and $conv4$ increases vertically and horizontally from zero to one. Notice how the details are different from one layer to another and how the combination of layers may result to more favored stylizations. For an interactive presentation please visit https://goo.gl/PVWQ9K.

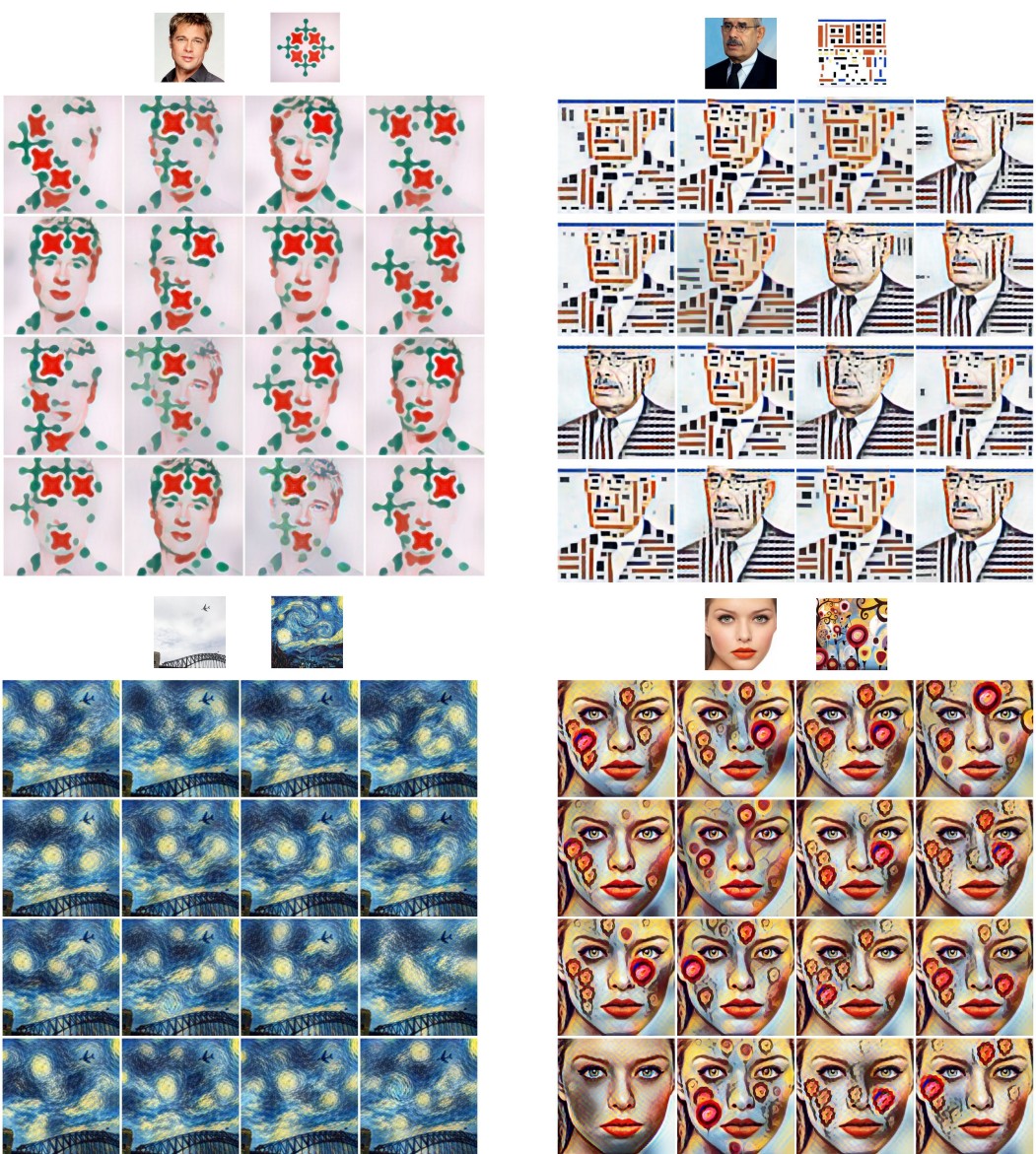

Figure 10: More results of stochastic stylization from the same pair of content/style. Each block represents randomized stylized outputs given the fix style/content image demonstrated at the top. Notice how stylized images vary in style granularity, the spatial position of style elements while maintaining similarity to the original style and content image. For more results please visit https://goo.gl/PVWQ9K.

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

| Operation | input dimensions | output dimensions |
|---|---|---|
| input parameters $\boldsymbol{\alpha}$ | 3 | 1000 |
| 10×Dense | 1000 | 1000 |
| Dense | 1000 | $2(\gamma_{\boldsymbol{\alpha}}, \beta_{\boldsymbol{\alpha}})$ |
| Optimizer | Adam ($\alpha = 0.001$, $\beta_1 = 0.9$, $\beta_2 = 0.999$) | |
| Training iterations | 200K | |
| Batch size | 8 | |
| Weight initialization | Isotropic gaussian ($\mu = 0$, $\sigma = 0.01$) | |

Table 1: Network architecture and hyper-parameters of $\Lambda$.

| Operation | Kernel size | Stride | Feature maps | Padding | Nonlinearity |
|---|---|---|---|---|---|
| **Network** – $256 \times 256 \times 3$ input | | | | | |
| Convolution | 9 | 1 | 32 | SAME | ReLU |
| Convolution | 3 | 2 | 64 | SAME | ReLU |
| Convolution | 3 | 2 | 128 | SAME | ReLU |
| Residual block | | | 128 | | |
| Residual block | | | 128 | | |
| Residual block | | | 128 | | |
| Residual block | | | 128 | | |
| Residual block | | | 128 | | |
| Residual block | | | 128 | | |
| Residual block | | | 128 | | |
| Upsampling | | | 64 | | |
| Upsampling | | | 32 | | |
| Convolution | 9 | 1 | 3 | SAME | Sigmoid |
| **Residual block** – $C$ feature maps | | | | | |
| Convolution | 3 | 1 | $C$ | SAME | ReLU |
| Convolution | 3 | 1 | $C$ | SAME | Linear |
| *Add the input and the output* | | | | | |
| **Upsampling** – $C$ feature maps | | | | | |
| *Nearest-neighbor interpolation, factor 2* | | | | | |
| Convolution | 3 | 1 | $C$ | SAME | ReLU |
| Normalization | Conditional instance normalization after every convolution | | | | |
| Optimizer | Adam ($\alpha = 0.001$, $\beta_1 = 0.9$, $\beta_2 = 0.999$) | | | | |
| Training iterations | 200K | | | | |
| Batch size | 8 | | | | |
| Weight initialization | Isotropic gaussian ($\mu = 0$, $\sigma = 0.01$) | | | | |

Table 2: Network architecture and hyper-parameters of $T$.

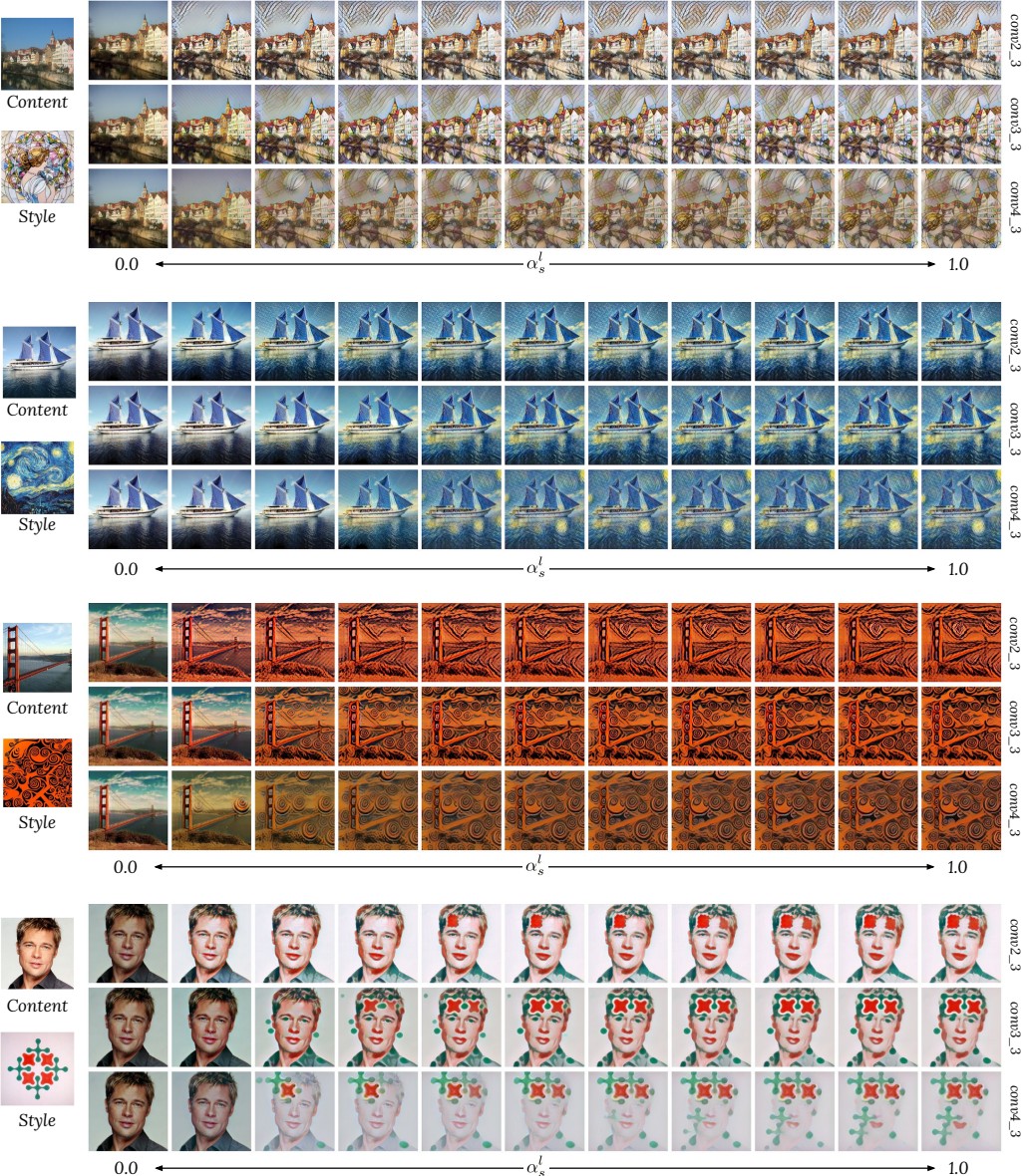

Figure 11: More examples for effect of adjusting the input parameters $\boldsymbol{\alpha}_s$ in real-time. Each row shows the stylized output when a single $\alpha_s^l$ increased gradually from zero to one while other $\boldsymbol{\alpha}_s$ are fixed to zero. Notice how the details of each stylization is different specially at the last column where the weight is maximum. Also how deeper layers use *bigger* features of style image to stylize the content.

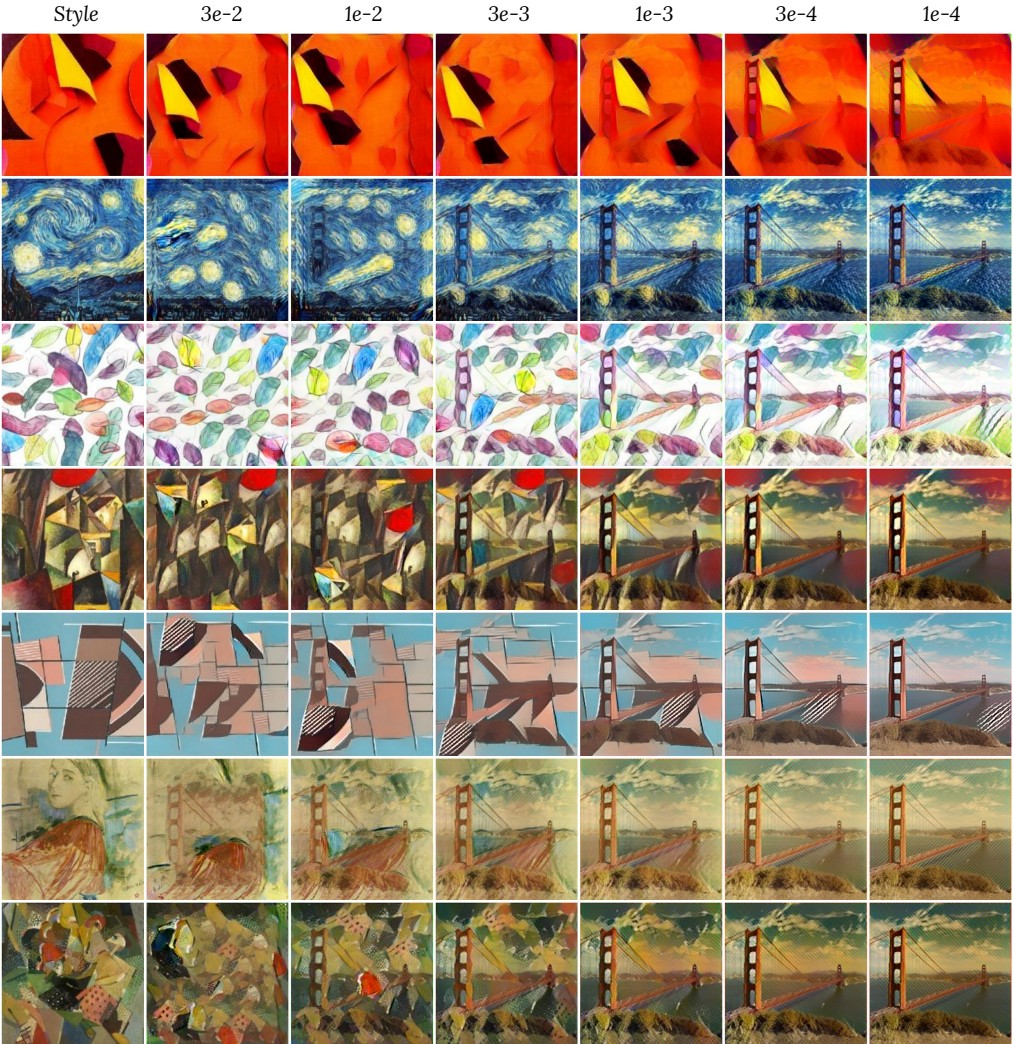

Figure 12: More examples for effect of adjusting the style weight in style transfer network from (Johnson et al., 2016). Each column demonstrates the result of a separate training. As can be seen, the "optimal" weight is different from one style image to another and there can be more than one "good" stylization depending on ones personal choice.

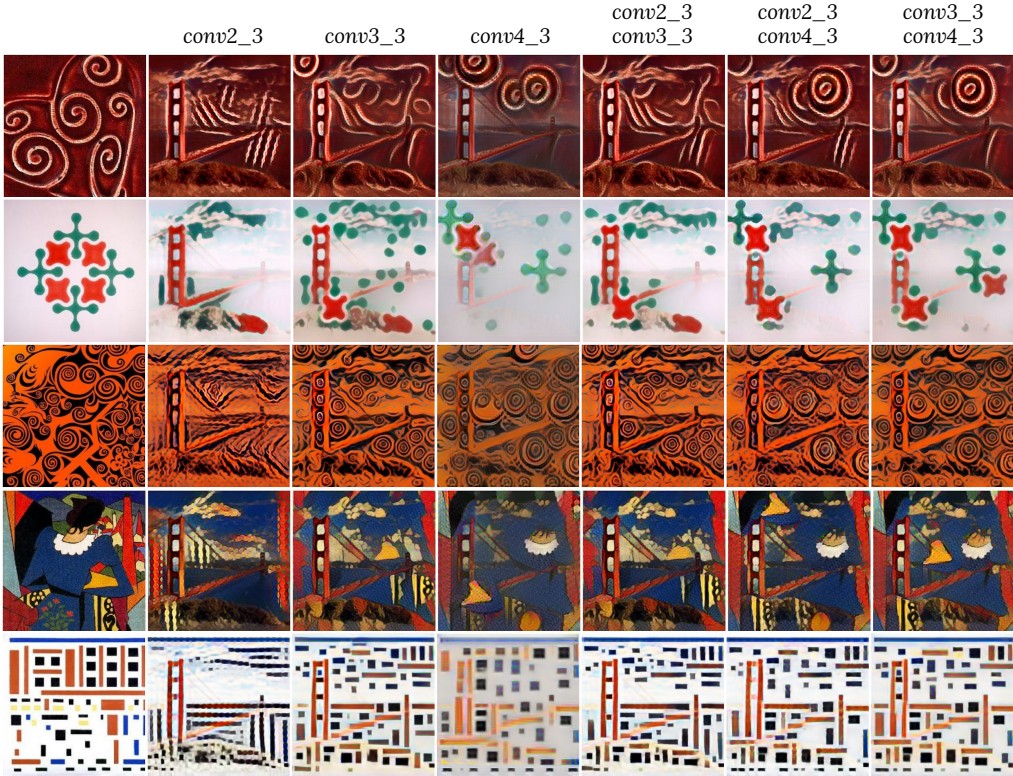

Figure 13: Results of combining losses from different layers at generation time by adjusting their corresponding parameters. The first column is the style image which is fixed for each row. The content image is the same for all of the outputs. The corresponding parameter for each one of the losses is zero except for the one(s) mentioned in the title of each column. Notice how each layer enforces a different type of stylization and how the combinations vary as well. Also note how a single combination of layers cannot be the "optimal" stylization for any style image and one may prefer the results from another column.

