# OpenReview forum: "Adjustable Real-time Style Transfer"
_ICLR.cc/2019/Workshop/DeepGenStruct — DeepGenStruct 2019_

### Official Review · AnonReviewer1 · 2019-04-15
**Interesting work, but lacking quantitative evaluation.**

**Rating:** 2
**Confidence:** 2

**Review:**

This paper presents an adjustable real-time style-transfer approach for generating a re-styled image given a style image and a content image, with the contribution that the stylisation of generated images could be adjusted/controlled at inference time by changing a few tuning parameters. Specifically, this is achieved by modeling as input the set of weights controlling the effect of the style and content captured by each layer of the network. The authors present various qualitative analysis to compare the proposed approach with existing works (StyleNet).

My major concern is the lack of quantitative experiments for evaluation. Specifically, how does the proposed approach contrast against existing models (e.g., StyleNet) in generating different stylizations with more diverse details (last paragraph, Section 2)? The authors only present one example in the experimental section, which seems to be insufficient.

Meanwhile, this submission only studies generating images, not other modalities with highly-structured representations, which might not fit the theme of this workshop.

Typos:
* Eq (1): should $\phi(\bm{s})$ be $\phi(\bm{c})$ in the first equation?
* Eq (5): format issue

---

### Official Review · AnonReviewer2 · 2019-04-17
**Review for "Adjustable Real-time Style Transfer"**

**Rating:** 3
**Confidence:** 1

**Review:**

I would like to prephase my review by saying that while I have reasonable experience in generative modeling, I know very little of style transfer beyond the basics, and this is the first paper I review in the topic, so my review cannot be more than an educated guess.

The paper at hand studies the problem of doing style transfer when the desired output is not just one image, but a collection of diverse images given a single style and content. Furthermore, the paper concentrates in real time generation (i.e. obtaining several images according to certain parameters without a need for retraining). This last part is where the novelty of the paper relies. To address this problem, the authors include as part of the network parameters alpha_c and alpha_s that are taken as an input for a neural network that produces as a feedforward pass the new images, and that's trained with images coming from weighted style transfer (eqs 2-3). The conditioning is done via instance normalization, very similar to [1].

The paper is very well written, the problem description and the algorithms (i.e. both contributions) are clear, and the method seems to 'perform well'. My main criticisms are that no quantitative evaluation or user studies have been done to assess the 'performance' of the model, and that there doesn't seem to be any fundamentally new ideas in play. Namely, it seems like a natural extension of existing methods. However, the ideas are well arranged and executed, and according to my judgement this merits enough for publication at a workshop like this one.

[1]: https://arxiv.org/pdf/1610.07629.pdf

---

### Decision · Program_Chairs · 2019-04-19
**Acceptance Decision**

**Decision:**

Accept

**Comment:**

This work presents a method for style transfer which enables users to modify the output via adjusting different hyperparameters. The reviewers agreed that this is a well-written paper with interesting experimental results.